# MST-SAM: Bridging Multi-View Gaps in SAM2 with Spatiotemporal Bank

## Abstract

High-quality, instance-level segmentations are crucial for developing multi-view vision-centric systems, such as self-driving vehicles and mobile robots, yet their annotation acquisition is prohibitively expensive. While human-in-loop labelling paradigms like SAM2 show great promise in monocular videos, adapting them to multi-cameras scenarios is hindered by two fundamental flaws: spatially, an ignorance of cross-view geometry leads to severe tracking ambiguity; and temporally, the exponential memory demands preclude real-time performance. To address these challenges, we propose **MST-SAM**, a novel streaming framework for robust, **m**ulti-view instance segmentation and tracking through **s**patio-**t**emporal bank. Our method introduces two core components: (1) a Spatio-Positional Augmentation (**SPA**) module that bridges SAM2's 2D-centric design with 3D scene geometry. It learns a unified positional prior from camera transformations, enabling tokens to reason about their absolute spatial location across different views. (2) a Memory View Selection (**MVS**) strategy that prunes the temporal memory bank, significantly reducing the computational overhead of the multi-view system while maintaining high algorithm performance. We validate our method on the nuScenes and Waymo datasets using a custom multi-view instance segmentation benchmark we introduce, where MST-SAM sets a new state of the art and demonstrates strong generalization.

## 1 Introduction

Recent advances in vision-centric systems, such as autonomous driving and mobile robotics, have underscored the critical need for high-quality, spatiotemporally consistent, instance-level annotations to support their perception models Lee et al. (2023); Ren et al. (2024); Cuttano et al. (2025). However, the high cost of dense multi-camera video labeling remains a bottleneck, driving research toward scalable, automated solutions.

Mainstream methods can be categorized into two pipelines. The first category of methods revolves around human-in-the-loop annotation pipelines built upon the Segment Anything Model 2 (SAM2) Kirillov et al. (2023); Ravi et al. (2024). Characterized by its online, low-cost, and highly interactive nature, this approach has emerged as a leading paradigm. It not only facilitates the efficient annotation of challenging "hard cases" but also allows users to perform fine-grained refinement on instance segmentation within a single frame via iterative prompting. However, its direct applicability to multi-camera systems remains limited Xu et al. (2025). The second category comprises modular approaches based on the BEV representation. These methods are centered on online detection and tracking algorithms, typically forming a cascaded pipeline of object detection, 3D tracking, and a SAM backend for segmentation. While these methods offer real-time capabilities, they often do so at the expense of accuracy and stability Tan et al. (2025); Li et al. (2022); Liu et al. (2023).

To leverage SAM2's zero-shot capabilities in multi-view settings, we first establish a baseline, **SAM2-MV**, using a simple Spatio-Temporal Bank that serializes multi-view inputs into a pseudo-sequence for compatibility with SAM2. the original SAM2 framework. Nevertheless, this straightforward solution suffers from fundamental design deficiencies, rendering it unsuitable for multi-camera contexts: Spatially, the model lacks robust 3D geometric priors Li et al. (2025), only relies on fragile 2D features and positional embeddings leads to severe tracking ambiguity for dense objects under ego-motion and fails to re-associate instances across abrupt viewpoint changes, causing

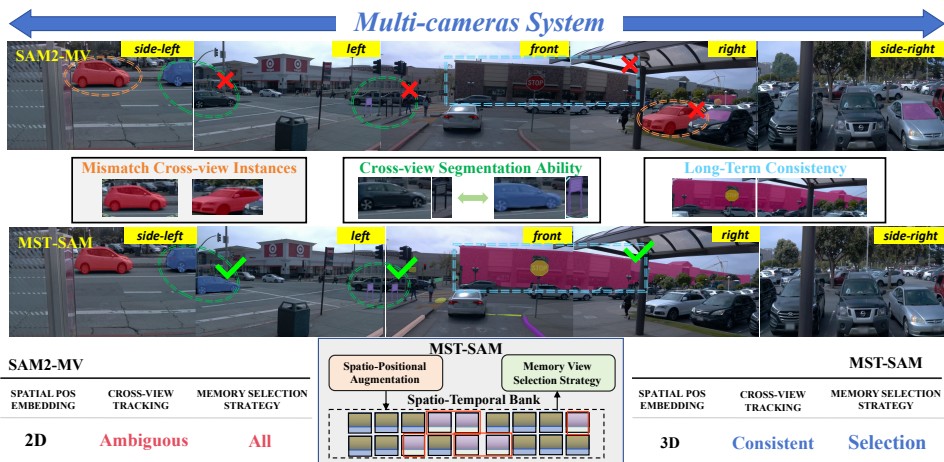

Figure 1: **MST-SAM**: To extend SAM2 for multi-camera systems, we propose MST-SAM, which introduces a Spatio-Positional Augmentation (SPA) module for geometric reasoning and Memory View Selection (MVS) strategy for efficient query. Building upon a baseline SAM2-MV (with a Spatio-Temporal Bank), MST-SAM achieves robust and consistent segmentation across multiple cameras, while maintaining a computational overhead comparable to the original SAM2.

frequent identity switches and tracking losses (Figure 1). Temporally, its memory design suffers from a combinatorial explosion in multi-camera streams. The memory bank's size, scaling with the number of views, imposes a prohibitive computational burden that renders real-time performance infeasible and introduces massive information redundancy Xu et al. (2025).

To overcome these limitations, we propose MST-SAM, a novel spatio-temporal strategy that enhance the naive Spatio-Temporal Bank, to achieve robust cross-view instance segmentation. This strategy is realized through two synergistic core components: (1) A Spatio-Positional Augmentation (SPA) module that injects crucial 3D geometric priors into the 2D feature space. This module leverages transformation matrices to establish a frustum projection between camera views. A lightweight, shared-weight network then encodes this geometric relationship into positional embeddings. By applying these embeddings to both memory and query tokens, the SPA module explicitly grounds them in a unified 3D space, enabling robust geometric reasoning and cross-view association. (2) A Memory View Selection (MVS) strategy that resolves the trade-off between temporal consistency and efficiency. Instead of naively accumulating features, the MVS assigns a dedicated memory token to each tracked instance. It then employs a selective update mechanism, which distills only the most salient information from recent frames for fusion via memory attention. This on-demand strategy ensures long-term temporal consistency while drastically reducing computational and storage overhead. The main contributions of this paper are:

- We propose MST-SAM, the first online framework to successfully adapt large segmentation models for robust instance tracking in multi-camera systems.
- Our Spatio-Positional Augmentation (SPA) module resolves cross-view ambiguity by pioneering a joint optimization of geometric priors and visual features at the feature level.
- We design a Memory View Selection (MVS) strategy that resolves the performance-efficiency trade-off in multi-camera streams.
- We establish a new multi-view instance segmentation benchmark where our method sets a new state-of-the-art (SOTA) on both nuScenes and Waymo datasets.

## 2 RELATED WORKS

### 2.1 OBJECT SEGMENTATION AND TRACKING IN VIDEOS

Video Object Segmentation (VOS) and its language-guided variant, Referring Video Object Segmentation (RVOS) Ravi et al. (2025); Liu et al. (2025); Qin et al. (2025), aim to track and segment

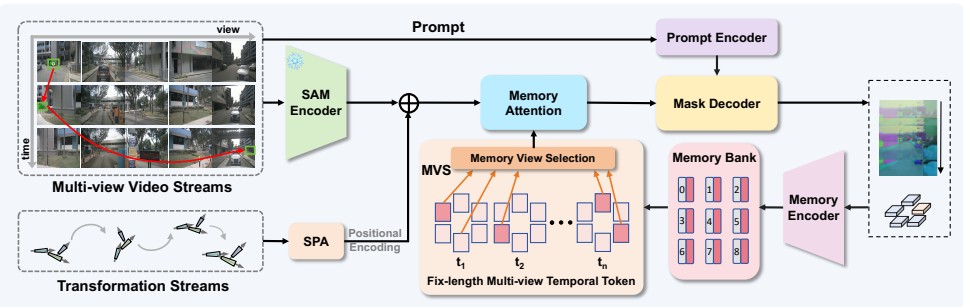

Figure 2: Architecture of the MST-SAM framework. The framework comprises two extra components upon SAM2-MV: the Spatio-Positional Augmentation (SPA) module and the Memory View Selection (MVS) strategy.

specified targets throughout a video sequence. Recent approaches have evolved from memory-based models that maintain temporal context to Transformer-based models that excel at object association.

Despite these advances, mainstream VOS methods are constrained by their reliance on 2D visual appearance cues Pont-Tuset et al. (2018); Oh et al. (2019); Voigtlaender et al. (2019). This reliance renders them fragile to common challenges like heavy occlusion and confusion from visually similar distractors Cheng & Schwing (2022); Cheng et al. (2021). Efforts to address these issues have led to a clear dichotomy: offline methods achieve high robustness by processing entire video sequences at once, but are unsuitable for real-time applications. Conversely, online methods prioritize low latency for streaming scenarios but are prone to error accumulation and identity drift. The recent advent of large-scale segmentation models like SAM2 offers a powerful new foundation Rajič et al. (2023); Kirillov et al. (2023). However, as models trained predominantly on static images, they inherit the same dependency on 2D appearance and do not resolve this long-standing trade-off between robustness and efficiency Yang et al. (2023); Ravi et al. (2024); Xu et al. (2025); Yang et al. (2021).

## 2.2 3D Instance-level Tracking

The dominant paradigm in 3D instance tracking is Tracking-by-Detection (TBD), which separates 3D detection from a subsequent association step Weng et al. (2020). The core challenge is data association, where current methods face a trade-off. Geometry-based metrics (e.g., 3D IoU) are robust but rely on expensive LiDAR data, while appearance-based features from cameras are detail-rich but prone to ambiguity Yin et al. (2021); Bai et al. (2022).

Consequently, state-of-the-art trackers often resort to LiDAR-camera fusion, which, despite its high performance, increases system cost and complexity. Camera-only trackers remain a more scalable alternative, but have historically struggled to match the robustness of fusion-based systems Kim et al. (2021); Pang et al. (2021); Wang et al. (2021). This highlights a critical need for methods that can integrate robust geometric reasoning directly into a camera-only framework, without the overhead of additional sensors or explicit 3D reconstruction.

## 3 Our Approach

### 3.1 Preliminary

Our approach addresses the task of prompt-guided, spatio-temporal 4D instance segmentation within vision-only, multi-view scenarios. Formally, given a multi-view video sequence of $T_v$ frames, and the associated vehicle ego-motion and sensor calibration data (intrinsics $K$ and extrinsics $E$) via poses, the task is initialized with a single point prompt $(i, t_0, w, h)$. This prompt identifies a target instance at coordinates $(w, h)$ in a specific camera view $i$ at a reference frame $t_0$. The objective is to predict a set of consistent binary masks, $s_t^i$, that segments the specified instance across all relevant views and subsequent frames.

To tackle this challenge, we propose MST-SAM, a framework designed to adapt the powerful, image-level Segment Anything Model 2 (SAM2) for this dynamic Ravi et al. (2024), multi-view

setting. As illustrated in Figure 2, our method is built upon two core innovations designed to be seamlessly integrated with a baseline SAM2 architecture.

The Spatio-Positional Augmentation (SPA) module is introduced to enforce spatio-temporal consistency. Its primary role is to provide robust geometric guidance for the segmentation model across different camera views and over time.

The Memory View Selection strategy is designed to enable real-time performance. It achieves this by intelligently managing computational resources and minimizing redundant processing.

### 3.2 FRAMEWORK OVERVIEW

The Segment Anything Model 2 (SAM2) is a potent architecture engineered for promptable object segmentation within a single video stream. It operates recursively, processing frames sequentially and maintaining temporal context via a dedicated memory mechanism. Its workflow at a given timestep $t$ can be summarized as follows:

**Feature Extraction (Encoder):** An image encoder, $\mathcal{E}_I$, independently extracts features $F_t$ from the current frame $I_t$ with formula: $F_t = \mathcal{E}_I(I_t)$.

**Memory Attention:** These features $F_t$ are fused with a memory bank $M$, which encapsulates the target's historical appearance, to yield memory-aware features $F_{\mathrm{mem},t}$ Ravi et al. (2024).

**Mask Decoder:** A mask decoder $\mathcal{D}_M$ then leverages the fused features $F\mathrm{mem}, t$ and an encoded prompt $E_P$ to predict the final segmentation mask $S_t$ Kirillov et al. (2023).

**Memory Encoder:** Finally, a memory encoder $\mathcal{E}_M$ updates the memory bank to $M_t$ using information from the current frame, preparing it for the next timestep. While highly effective for single videos, SAM2's design is inherently uni-stream. Its memory mechanism propagates information temporally along a single camera's timeline, making it ill-suited for multi-view systems.

Our proposed MST-SAM model introduces a naive adaptation strategy, the **Spatio-Temporal Bank**, which serializes the multi-view video stream $(I_t^1, I_t^2, \ldots, I_t^N, I_{t+1}^1, \ldots)$ into a single pseudo-sequence. This enables seamless integration with the native SAM2 architecture, allowing core modules like the memory attention and memory encoder to be reused without modification. We designate this baseline model as **SAM2-MV**. Despite being the most direct approach, it is critically flawed and introduces two fundamental challenges:

Although this appears to be the most straightforward approach to retain SAM2's core architecture, it is critically flawed and introduces two fundamental challenges:

- **Spatio-Temporal Ambiguity:** The model loses geometric context, becoming unable to distinguish a temporal step (from $t$ to $t+1$) from a viewpoint shift (from camera $i$ to $i+1$). This leads to inconsistent tracking across views.

- **Computational Inefficiency:** The memory sequence length is multiplied by the number of views ($N$), drastically increasing the computational load of the memory mechanism and processing redundant information from overlapping camera perspectives Ravi et al. (2024).

Addressing these specific challenges is the primary motivation for our work. The following sections detail the modules we designed to resolve this ambiguity and inefficiency.

### 3.3 SPATIO-POSITIONAL AUGMENTATION

To resolve the spatio-temporal ambiguity inherent in serialized multi-view streams, we introduce the Spatio-Positional Augmentation (SPA) module. The overall process, illustrated in Figure 5, is designed to endow the model with explicit geometric awareness. It achieves this by generating and fusing two distinct forms of positional information: (1) It retains the standard 2D positional encoding $P_{\mathrm{img}}$, which captures the relative spatial structure within a single frame. (2) Crucially, it introduces a simple but novel frustum-based geometric encoding $P_{\mathrm{3d}}$, which is generated by lifting 2D pixel coordinates into a unified 3D space using the camera's intrinsic and extrinsic parameters.

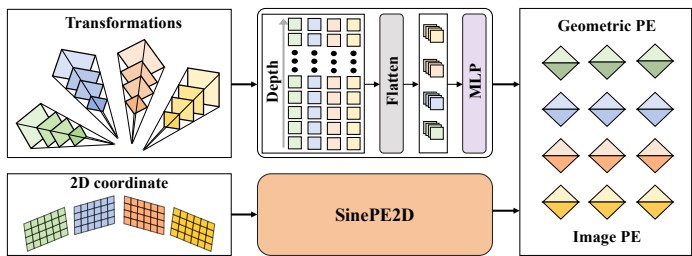

Figure 3: The Spatio-Positional Augmentation (SPA) generates a 3D-aware Geometric PE (positional embedding) using camera parameters and fuses it with a standard 2D Image PE.

By combining these two encodings, SPA provides each feature token with a rich positional prior that understands both its location on the 2D image plane and its origin within the 3D world, thereby directly addressing the ambiguity between viewpoint shifts and temporal progression.

The necessity for our SPA module stems from the fundamental limitations of the standard 2D positional encoding, $P_{\text{img}}$. While effective for providing relative position information within a single frame, this approach has a critical flaw in a multi-camera context: it is inherently agnostic to the camera's extrinsic and intrinsic parameters. Consequently, a pixel at coordinate $(u, v)$ in the front camera and a pixel at the same coordinate $(u, v)$ in a side camera are assigned identical positional values. This ambiguity makes it impossible for the model to differentiate between a genuine change in the scene over time and a simple switch in camera viewpoint. To break this symmetry and resolve the ambiguity, we must introduce a positional encoding that is aware of 3D geometry.

Our solution, the frustum-based geometric encoding $P_{\text{3d}}$, directly generates this 3D-aware signal. As depicted in the top path of Figure 3, the core mechanism is Frustum Lifting. This process lifts 2D pixels from any source view (be it the current frame or a historical memory frame) into the unified 3D coordinate system of the current base frame.

Specifically, for each pixel $(u, v)$, we hypothesize its location along a viewing ray by sampling a set of $N$ discrete depths, $D = d_1, d_2, \ldots, d_N$. Each pixel-depth pair is then back-projected from the source camera's 2D image plane into a 3D point in the base frame's coordinate system. The transformation for a single depth hypothesis $d_i$ is formulated as:

$$\mathbf{x}_{u,v,d_i} = \mathbf{T}_{\text{base}\leftarrow\text{src}} \cdot \mathbf{K}_{\text{src}}^{-1} \cdot [u \cdot d_i, v \cdot d_i, d_i, 1]^T \tag{1}$$

In Equation 1, $\mathbf{K}_{\text{src}}$ is the $4 \times 4$ homogeneous intrinsic matrix of the source camera. The transformation $\mathbf{T}_{\text{base}\leftarrow\text{src}}$ is critical, as it unifies points from different cameras and different timestamps into a single reference frame: the ego-actor's coordinate system at the current time $t_{\text{base}}$. This composite transformation elegantly handles both spatial and temporal alignment by chaining two operations:

Camera-to-Ego Transformation $\mathbf{T}_{\text{ego}(t_{\text{src}}\leftarrow\text{cam}(t_{\text{src}}))}$, the camera's extrinsic matrix, which transforms the point from the source camera's local coordinate system to the ego-actor's coordinate system at that same source timestamp, $t_{\text{src}}$. Ego-Motion Transformation $\mathbf{T}_{\text{ego}(t_{\text{base}})\leftarrow\text{ego}(t_{\text{src}})}$, this matrix accounts for the vehicle's movement between the source's timestamp $t_{\text{src}}$ and the current base timestamp $t_{\text{base}}$. The complete transformation is thus formulated as:

$$\mathbf{T}_{\text{base}\leftarrow\text{src}} = \mathbf{T}_{\text{ego}(t_{\text{base}})\leftarrow\text{ego}(t_{\text{src}})} \cdot \mathbf{T}_{\text{ego}(t_{\text{src}})\leftarrow\text{cam}(t_{\text{src}})} \tag{2}$$

This formulation ensures that whether a pixel comes from a side camera in the current frame or the front camera from a past frame, its resulting 3D points are all expressed relative to the ego's current position and orientation.

With all 3D points $\mathbf{x}_{u,v,d_i}$ unified in the base frame, we distill this information into a single feature vector for each pixel. This is a two-stage process of point-wise encoding followed by aggregation:

First, to create a feature for each point on the viewing ray, we concatenate the 3D coordinates $\mathbf{x}_{u,v,d_i}$ with their corresponding depth value $d_i$. This combined vector is processed by a lightweight MLP to produce a point-wise geometric encoding, $P_{\text{3d}}^{(i)}$. This allows the model to learn features sensitive

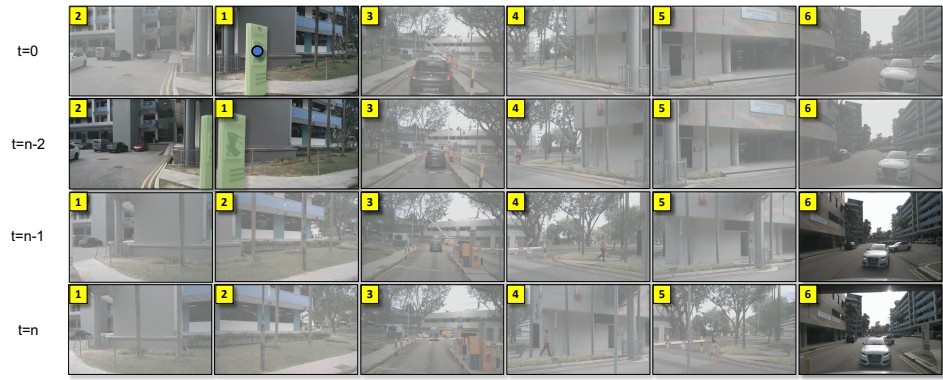

Figure 4: Visualization of our **Memory View Selection (MVS) strategy** in action. The filter prunes irrelevant historical views (faded) while selecting high-quality features (bright, with masks) and the initial anchor frame ($t = 0$) to build a compact and effective memory bank.

to both the absolute position in space and the distance from the camera.

$$P_{3d}^{(i)}(u, v) = \text{MLP}(\text{Concat}(\mathbf{x}_{u,v,d_i}, d_i)) \tag{3}$$

Next, the set of $N$ point-wise encodings, collectively describing the geometry of the pixel's viewing ray, is aggregated into a single unified geometric encoding $P_{3d}$ using an aggregation function like max-pooling, which selects the most salient geometric features along the ray:

$$P_{3d}(u, v) = \underset{i=1,\dots,N}{\text{Aggregate}} \left( P_{3d}^{(i)}(u, v) \right) \tag{4}$$

Finally, to complete the Spatio-Positional Augmentation (SPA), this learned geometric encoding $P_{3d}$ is fused with the standard 2D sinusoidal encoding $P_{\text{img}}$ via element-wise addition.

$$P_{\text{SPA}}(u, v) = P_{\text{img}}(u, v) + P_{3d}(u, v) \tag{5}$$

The resulting fused encoding, $P_{\text{SPA}}$, is then applied to all image and memory tokens. By enriching the original 2D positional information with a learned, 3D-aware geometric prior, this process effectively resolves the spatio-temporal ambiguity, making the model fully aware of the underlying camera geometry and temporal vehicle motion.

### 3.4 MEMORY VIEW SELECTION STRATEGY

To manage the prohibitive memory and computational costs of processing long video streams, we introduce the **Memory View Selection strategy**. As shown in Figure 4, this module intelligently prunes the memory bank based on the principle of **Spatio-Temporal Continuity**: assuming relevant information is concentrated in recent, high-quality views.

The filtering process is executed at each inference step, governed by a set of prioritized rules:

**Anchor Frame:** The features from the initial "conditioning frame" are always retained, serving as a stable anchor of the target's appearance to prevent long-term drift.

**Prioritized Selection:** For all other historical frames, we apply a selection hierarchy:

- **Mask-First:** We prioritize features from views that produced a valid segmentation mask, as a mask is a strong positive signal of the target's presence.
- **Same-Camera Fallback:** If no view at a timestamp has a valid mask, we fall back to selecting the feature from the *same camera source* as the current view, leveraging spatial continuity to maximize view overlap.

**Recency Pruning & Assembly:** Any historical feature outside a predefined temporal window that consistently fails the above criteria is discarded to prevent stale information from accumulating.

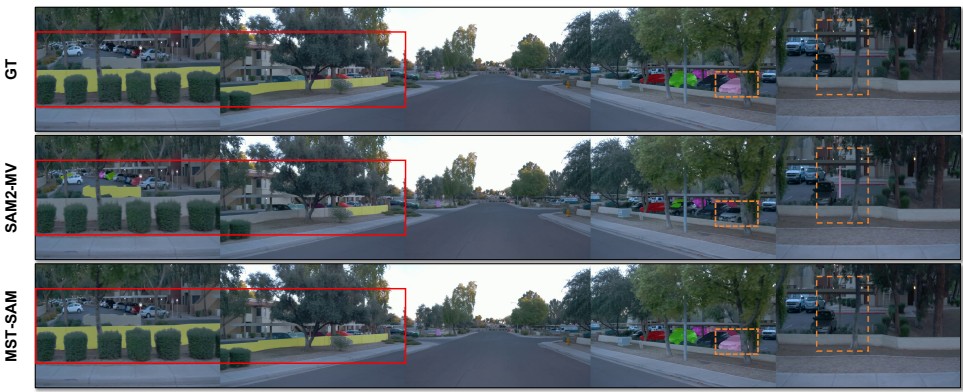

Figure 5: Qualitative comparison of segmentation with SAM2-MV on the Waymo Open dataset. Different colors represent different objects. The displayed image has been resized for better visual effects. Where the red boxes denote identity confusion and orange boxes denote identity mismatch.

From the remaining filtered pool, the $k$ most recent features are selected, combined with the anchor frame, and fed to the Memory Attention module.

This strategy is designed as a practical instantiation of the Information Bottleneck principle for temporal processing. It dynamically distills the redundant spatio-temporal stream into a compact yet potent memory bank, seeking to retain only the information maximally relevant to the core task of long-term identity preservation. The resulting representation effectively balances informational richness with computational efficiency. Our subsequent ablation studies, where we analyze various filtering criteria, empirically confirm that this principled distillation approach achieves a superior accuracy-efficiency trade-off (Table 2).

### 3.5 LOSS FUNCTION

We adopt the established training objective and robust optimization strategy from SAM-2 to ensure a fair comparison. The total loss, $\mathcal{L}_{\text{total}}$, is a linear combination of three distinct losses: mask prediction, IoU prediction, and object presence prediction Lin et al. (2018); Milletari et al. (2016):

$$\mathcal{L}_{\text{total}} = \mathcal{L}_{\text{mask}} + \mathcal{L}_{\text{IoU}} + \mathcal{L}_{\text{presence}} \tag{6}$$

where $\mathcal{L}_{\text{mask}}$ is the mask supervision loss, defined as a weighted sum of Focal Loss ($\mathcal{L}_{\text{focal}}$) and Dice Loss ($\mathcal{L}_{\text{dice}}$): $\mathcal{L}_{\text{mask}} = \lambda_{\text{focal}}\mathcal{L}_{\text{focal}} + \mathcal{L}_{\text{dice}}$. $\mathcal{L}_{\text{IoU}}$ is the MAE loss for the predicted IoU score, and $\mathcal{L}_{\text{presence}}$ is the Cross-Entropy loss for object presence prediction.

## 4 EXPERIMENTS

### 4.1 DATASETS AND BENCHMARKS

We conduct a comprehensive evaluation of our method on two benchmarks derived from large-scale autonomous driving datasets: **nuScenes** Caesar et al. (2020) and the **Waymo Open Dataset (WOD)** Sun et al. (2020). Both are sourced from multi-camera systems and are characterized by an abundance of dynamic instances. To ensure a thorough and fair evaluation, we categorize instances based on their visibility across camera views into two distinct types: (1) **Cross-view Objects:** Instances that appear sequentially across multiple camera views. (2) **Single-view Objects:** Instances that are captured exclusively within a single camera's field of view throughout the sequence. More details about Waymo open dataset and nuScenes dataset are shown in the Appendix.

### 4.2 IMPLEMENTATION DETAILS AND BASELINES

Our model is initialized with pre-trained weights from the official SAM2 release. To adapt it to multi-view tracking, we follow the overall fine-tuning protocol of SAM2, updating all components

except the memory encoder, which remains frozen during training. This preserves the rich pre-trained visual representations while enabling the model to learn cross-view associations.

We train using a hybrid sampling strategy on Waymo and nuScenes datasets, with 80% cross-view instances (encouraging multi-view consistency) and 20% single-view instances (to regularize training and maintain single-view performance). We use the AdamW optimizer, a cosine learning rate schedule starting at $5 \times 10^{-6}$, and train for 12 epochs.

We evaluate our model against several baselines, **SAM2-MV**: This variant equips SAM2 with a spatio-temporal bank strategy, utilizing a memory bank window of size 8; **MST-SAM-L**: A lightweight variant of our model. It removes the Memory View Selection (MVS) strategy and the Ego-Motion Transformation component from the Spatial Prompt Aggregation (SPA) module. Consequently, spatial aggregation is confined to the current frame. This version utilizes a memory bank window of size 18; **MST-SAM-G**: A simplified variant that removes the MVS strategy but retains the complete SPA module. This configuration is designed to isolate and evaluate the contribution of geometric-aware prompt aggregation. It uses a memory bank window of size 18; **MST-SAM-M**: A variant where the MVS module is streamlined to incorporate only the anchor frame mechanism and the mask-first prompting strategy. The memory bank window size is set to 8; **MST-SAM**: Our full proposed model, which integrates all components including the complete MVS and SPA modules. It operates with a memory bank window of size 8; **FastPoly-SAM**: A two-stage baseline that first performs 3D tracking with FastPoly Li et al. (2024) and then projects the tracks onto 2D views to prompt a SAM Kirillov et al. (2023) backend for dense segmentation. Further architectural details are available in the Appendix.

## 4.3 QUANTITATIVE EXPERIMENTS

Table 1: Main quantitative results on **cross-view benchmark**. A.2

| Method | Waymo | | | nuScenes | | | Latency (ms) ↓ |
|---|---|---|---|---|---|---|---|
| | $J\&F \uparrow$ | $J \uparrow$ | $F \uparrow$ | $J\&F \uparrow$ | $J \uparrow$ | $F \uparrow$ | |
| SAM2-MV | 84.92 | 84.23 | 85.62 | 92.07 | 91.67 | 92.46 | 72.1 |
| Fast-Poly-SAM | 71.22 | 69.14 | 73.31 | 74.11 | 75.32 | 72.89 | **47.2** |
| MST-SAM-G | 91.71 | 90.89 | 92.51 | 92.73 | 92.20 | 93.26 | 233.5 |
| MST-SAM-M | 90.39 | 89.57 | 91.21 | 91.88 | 91.22 | 92.54 | 73.2 |
| MST-SAM | **91.78** | **90.91** | **92.64** | **93.18** | **92.56** | **93.80** | 71.2 |

As shown in Table 1, we present a comprehensive quantitative evaluation of our method on the Waymo and nuScenes datasets. Our analysis covers overall performance, individual component contributions, comparisons with alternative methods, and the accuracy-efficiency trade-off.

Our full model, MST-SAM, demonstrates clear superiority, achieving the highest performance across all evaluated models. On the challenging Waymo dataset, it obtains a $J\&F$ score of **91.78**. This marks a substantial **+6.86** improvement over the powerful SAM2 baseline (84.92), validating the significant benefits of our integrated multi-view tracking and segmentation architecture.

To dissect the contribution of each proposed module, we analyze the performance progression across our internal variants. The leap from MST-SAM-G to MST-SAM (91.71 → 91.78) underscores the contribution of the MVS strategy in fusing temporal information. It is crucial to note that MST-SAM achieves this superiority despite using a much smaller memory window (8 vs. 18 for the variants), which further emphasizes the efficiency and effectiveness of our temporal fusion strategy. We further compare our integrated model against FastPoly-SAM, a baseline representing a classic cascaded ("track-in-3D-then-segment-in-2D") paradigm. While FastPoly-SAM achieves a competitive score of 71.22, our MST-SAM still outperforms it. We attribute this to our end-to-end architecture, which jointly optimizes perception and association. This integrated design effectively mitigates the **error propagation** issues inherent in pipeline approaches, where inaccuracies from the 3D tracker can directly degrade the quality of the subsequent segmentation.

Beyond accuracy, our method strikes an excellent balance with efficiency. MST-SAM delivers its state-of-the-art performance at a practical latency of only **71.2 ms**. For applications with stricter real-time constraints, the MST-SAM-M variant offers a compelling alternative.

Table 2: Quantitative evaluation and zero-shot generalization on the **cross-view benchmark** A.2. All models are trained on a single dataset to assess in-domain performance and cross-dataset generalization. A → B means training on the A dataset and evaluating on the B dataset.

| Method | Waymo → Waymo | | | Waymo → nuScenes | | | nuScenes → Waymo | | | nuScenes → nuScenes | | | latency |
|---|---|---|---|---|---|---|---|---|---|---|---|---|---|
| | $J\&F\uparrow$ | $J\uparrow$ | $F\uparrow$ | $J\&F\uparrow$ | $J\uparrow$ | $F\uparrow$ | $J\&F\uparrow$ | $J\uparrow$ | $F\uparrow$ | $J\&F\uparrow$ | $J\uparrow$ | $F\uparrow$ | (ms)$\downarrow$ |
| SAM2-MV | 84.92 | 84.23 | 85.62 | 92.07 | 91.67 | 92.46 | 84.92 | 84.23 | 85.62 | 92.07 | 91.67 | 92.46 | 72.1 |
| MST-SAM-L | **92.46** | **91.70** | **93.18** | 93.85 | 93.39 | 94.32 | 92.03 | 91.23 | **92.03** | 94.18 | 93.48 | **94.87** | 233.1 |
| MST-SAM-G | 92.01 | 91.18 | 92.83 | **94.01** | **93.45** | **94.57** | **92.04** | **93.30** | 91.65 | 93.00 | 93.48 | 93.53 | 233.5 |
| MST-SAM-M | 90.14 | 89.38 | 90.90 | 93.49 | 92.86 | 94.12 | 89.74 | 88.85 | 90.64 | 93.61 | 93.05 | 94.18 | 73.2 |
| MST-SAM | 91.06 | 90.36 | 91.76 | 93.39 | 92.85 | 93.94 | 90.61 | 89.66 | 91.57 | 92.48 | 91.66 | 93.31 | **71.2** |

Table 3: Quantitative evaluation and zero-shot generalization on the **generic object benchmark** A.2.

| Method | Waymo → Waymo | | | Waymo → nuScenes | | | nuScenes → Waymo | | | nuScenes → nuScenes | | |
|---|---|---|---|---|---|---|---|---|---|---|---|---|
| | $J\&F\uparrow$ | $J\uparrow$ | $F\uparrow$ | $J\&F\uparrow$ | $J\uparrow$ | $F\uparrow$ | $J\&F\uparrow$ | $J\uparrow$ | $F\uparrow$ | $J\&F\uparrow$ | $J\uparrow$ | $F\uparrow$ |
| SAM2-MV | 84.90 | 84.35 | 85.50 | 91.70 | 91.21 | 92.20 | 84.90 | **84.35** | 85.50 | 91.70 | 91.21 | 92.20 |
| MST-SAM-L | 85.60 | 84.64 | 86.56 | 92.14 | 91.49 | 92.80 | 84.60 | 84.08 | 85.24 | 93.23 | 92.60 | 93.85 |
| MST-SAM-G | **86.13** | **85.45** | **86.80** | 92.42 | 91.89 | 92.95 | **84.92** | 84.23 | **85.62** | 92.90 | 92.30 | 93.41 |
| MST-SAM-M | 83.73 | 83.20 | 84.26 | 91.53 | 90.94 | 92.11 | 81.55 | 80.88 | 82.24 | **93.54** | 92.53 | **94.04** |
| MST-SAM | 84.91 | 84.38 | 85.43 | **92.72** | **92.13** | **93.31** | 83.77 | 83.20 | 84.34 | 93.45 | **92.91** | 93.99 |

## 4.4 ABLATION STUDIES

Table 2 and Table 3 present our detailed quantitative analysis, which validates the effectiveness of our proposed modules and their ability to preserve generalization.

**Effectiveness of SPA.** The efficacy of our Spatio-Positional Augmentation (SPA) module is immediately evident from Table 2. Even the local augmentation pipline MST-SAM-L outperforms the SAM2 baseline, improving the J&F score from 84.92 to 92.46 on the Waymo→Waymo task. This confirms that introducing geometric priors via SPA is crucial for adapting SAM2's powerful segmentation capabilities to the multi-view domain.

**Efficiency and Intelligence of MVS.** The MVS strategy is designed to resolve the accuracy-efficiency trade-off. While a model with a full memory bank like MST-SAM-G suffers from prohibitive latency 233.5 ms, our full model, MST-SAM, equipped with MVS, operates at a real-time latency of 71.2 ms. Crucially, this efficiency is not achieved by naive memory truncation; MST-SAM significantly outperforms the MST-SAM baseline (91.06 vs. 90.14 J&F). This demonstrates that MVS intelligently filters the memory bank, preserving critical information and achieving a near-optimal balance between performance and speed.

**Preservation of Generalization Ability.** A key concern is whether our multi-view adaptations degrade the model's inherent generalization. Table 3 demonstrates this is not the case. The performance of our MST-SAM on generic (non-cross-view) object benchmark remains on par with the original SAM2, confirming our modules do not cause catastrophic forgetting. Interestingly, variants with larger memory banks even show a slight performance boost. This highlights the value of a rich memory context and reinforces the importance of our memory view selection strategy, which efficiently manages this valuable information without high computational costs.

## 5 CONCLUSION

In this paper, we proposed MST-SAM, a novel online framework to overcome the spatial ambiguity and prohibitive memory costs of adapting single-view segmentation models to multi-view streaming environments. Our method introduces a Spatio-Positional Augmentation (SPA) module to inject geometric priors for robust cross-view tracking and a Memory View Selection (MVS) strategy to ensure computational efficiency. Extensive experiments on nuScenes and Waymo show that MST-SAM sets a new state of the art on our proposed benchmark, demonstrating a practical path toward scalable semi-automated annotation and robust online perception for autonomous systems.

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

# A  APPENDIX

## A.1  MORE DETAILS OF TRAINING DETAILS

We follow the overall fine-tuning setup of SAM2, adopting a parameter-efficient fine-tuning (PEFT) strategy to adapt the foundation model to our cross-view task. All models are initialized from the official SAM2 checkpoints. During training, we freeze the memory encoder while allowing gradients to propagate through all other components, including the image encoder, prompt encoder, and mask decoder. This approach strikes a balance between training efficiency and effective adaptation: by keeping the memory encoder fixed, we preserve the temporal consistency modeling capabilities learned during pre-training, while enabling the rest of the network to adjust to the new task domain.

To train our model effectively, we introduce a hybrid data sampling strategy designed to balance the learning of cross-view consistency with single-view generalization. In each training batch, we combine instances from two sources:

- **Cross-view Instances (80% of batch):** These are object instances tracked across multiple synchronized camera views. This data is pivotal for training the model to learn robust feature associations and maintain consistent identity representations despite significant viewpoint changes. They directly address the core challenge of cross-view segmentation.
- **Single-view Instances (20% of batch):** These are instances that appear in only a single camera's field of view. Their inclusion acts as a regularization mechanism, preventing the model from overfitting to the potentially sparse multi-view data. Furthermore, it ensures the model retains strong segmentation performance in scenarios involving heavy occlusion or when objects are only partially visible, enhancing its overall robustness.

## A.2  EVALUATION METRICS AND BENCHMARKS

To provide a comprehensive and multi-faceted evaluation of segmentation quality, we adopt two widely-recognized metrics that assess both region-level accuracy and boundary-level precision.

- **Region Similarity** ($\mathcal{J}$), commonly known as the Jaccard Index or Intersection over Union (IoU), quantifies the spatial overlap between the predicted mask ($M$) and the ground-truth mask ($G$). It provides a holistic measure of how well the predicted region corresponds to the actual object area and is calculated as:

$$\mathcal{J} = \frac{|M \cap G|}{|M \cup G|}$$

- **Contour Accuracy** ($\mathcal{F}$) serves as a complementary metric that focuses specifically on the quality of the segmentation boundary. It is defined as the F-measure, which is the harmonic mean of precision and recall calculated over the boundary pixels. This metric is crucial for evaluating the model's ability to delineate fine details and is computed as:

$$\mathcal{F} = \frac{2 \cdot \text{Precision} \cdot \text{Recall}}{\text{Precision} + \text{Recall}}$$

Finally, to provide a single, unified score for overall comparison and ranking, we report the **Overall Performance** ($\mathcal{J}\&\mathcal{F}$), which is the arithmetic mean of the $\mathcal{J}$ and $\mathcal{F}$ scores.

To comprehensively assess the performance of MST-SAM, we establish two distinct evaluation benchmarks: the Cross-view Benchmark and the Comprehensive Benchmark.:

- **Cross-view Benchmark**, this benchmark is specifically designed for the instance segmentation task on multi-camera datasets. For evaluation, it exclusively considers instances that appear in two or more camera views throughout the temporal sequence. This benchmark aims to specifically measure the model's ability to associate and segment the same object across different perspectives.
- **Generic object benchmark**, this benchmark evaluates the overall instance segmentation performance on multi-camera datasets. The evaluation scope includes all instances that have appeared at any point in any camera view within the entire temporal sequence, with no restriction on the number of views.

### A.3 Modular Baseline

The FastPoly-SAM method Li et al. (2024), which we reference in the main text, represents a modular, pipelined framework for multi-view instance segmentation, also enabling the annotation of specific target instances across views. As depicted in Figure 6-(a), its workflow is composed of several distinct stages. Initially, a detector processes the raw sensor inputs to generate bounding boxes. Subsequently, in the initial frame, a filtering step selects the target instance IDs designated for segmentation. For all subsequent frames, a tracking model maintains the instance trajectories. These tracks are then projected onto the 2D image planes using camera transformation matrices. Finally, the resulting 2D bounding boxes serve as prompts for a downstream SAM module, which executes the final segmentation.

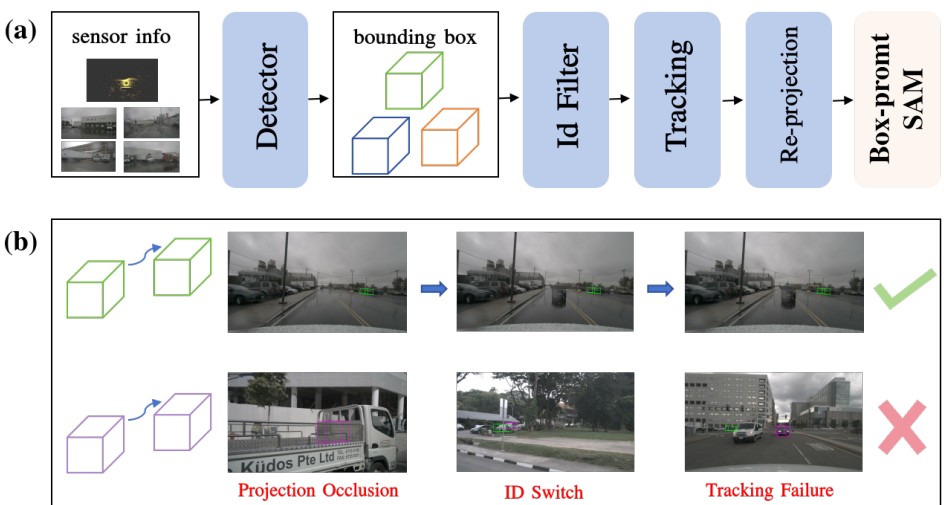

Figure 6: Analysis of the Modular Baseline Architecture and its Performance.

However, as illustrated in Figure 6-(b) and quantitatively demonstrated in Table 1, such a cascaded paradigm suffers from fundamental limitations. These include critical failure modes such as Projection Occlusion, Identity Switches, and complete Tracking Failures. Consequently, the method exhibits a significant performance deficit on key evaluation metrics. We attribute this inferiority to the fact that its explicit, yet rigid, utilization of 3D information is insufficient to meet the complex demands of practical annotation tasks.

### A.4 More visualization

we provide more visualization results at Figure 7, Figure 8, Figure 9, and Figure 10.

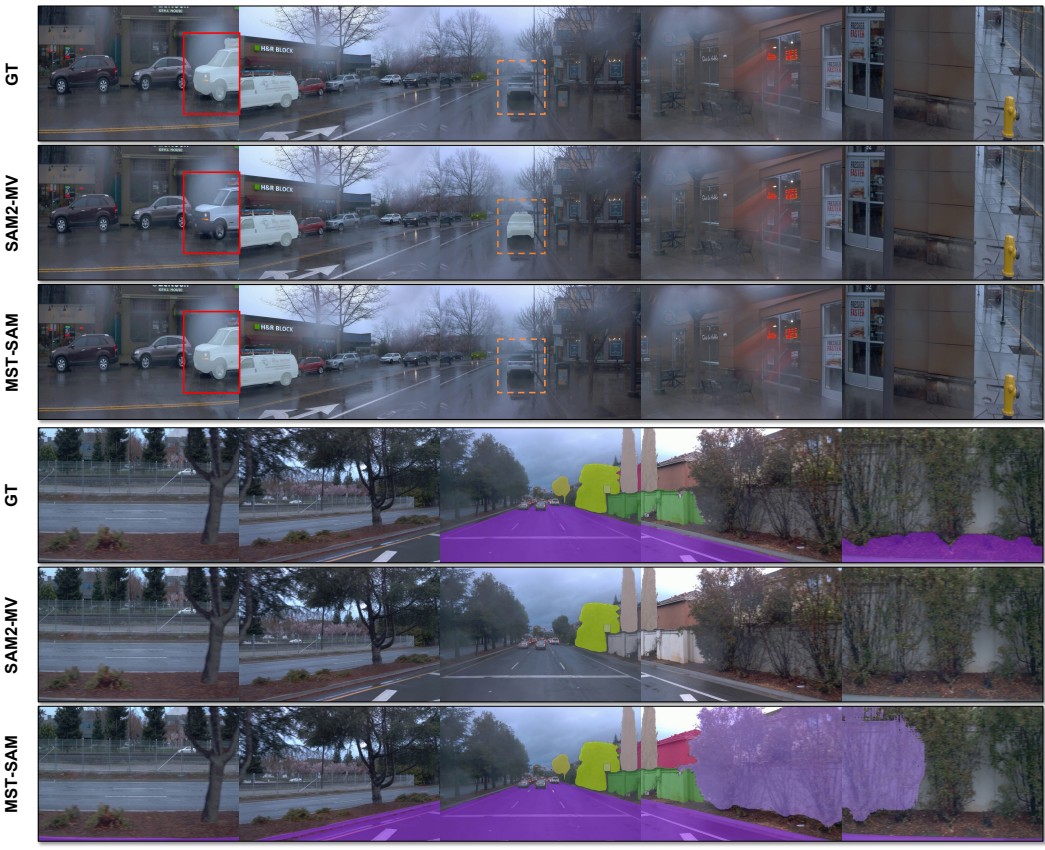

Figure 7: More qualitative comparison of segmentation with SAM2-MV on the Waymo Open dataset. Different colors represent different objects. The displayed image has been resized for better visual effects. The results of MST-SAM outperform the ground truth annotated with Xu et al. (2025).

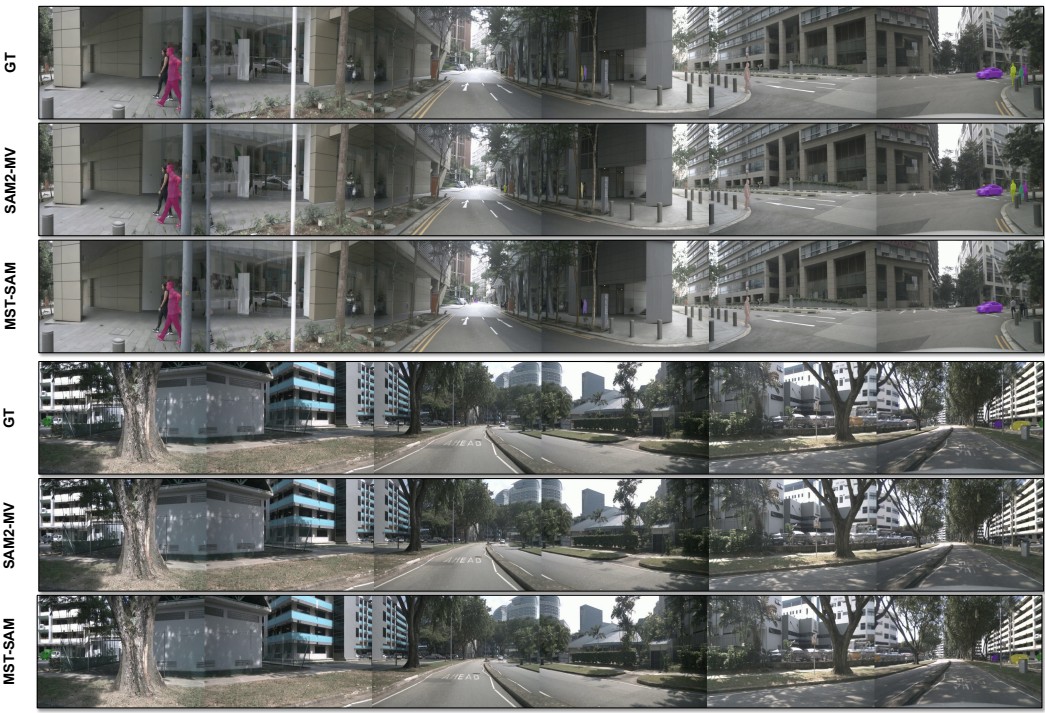

Figure 8: More qualitative comparison of segmentation with SAM2-MV on the nuScenes dataset. Different colors represent different objects. The displayed image has been resized for better visual effects.

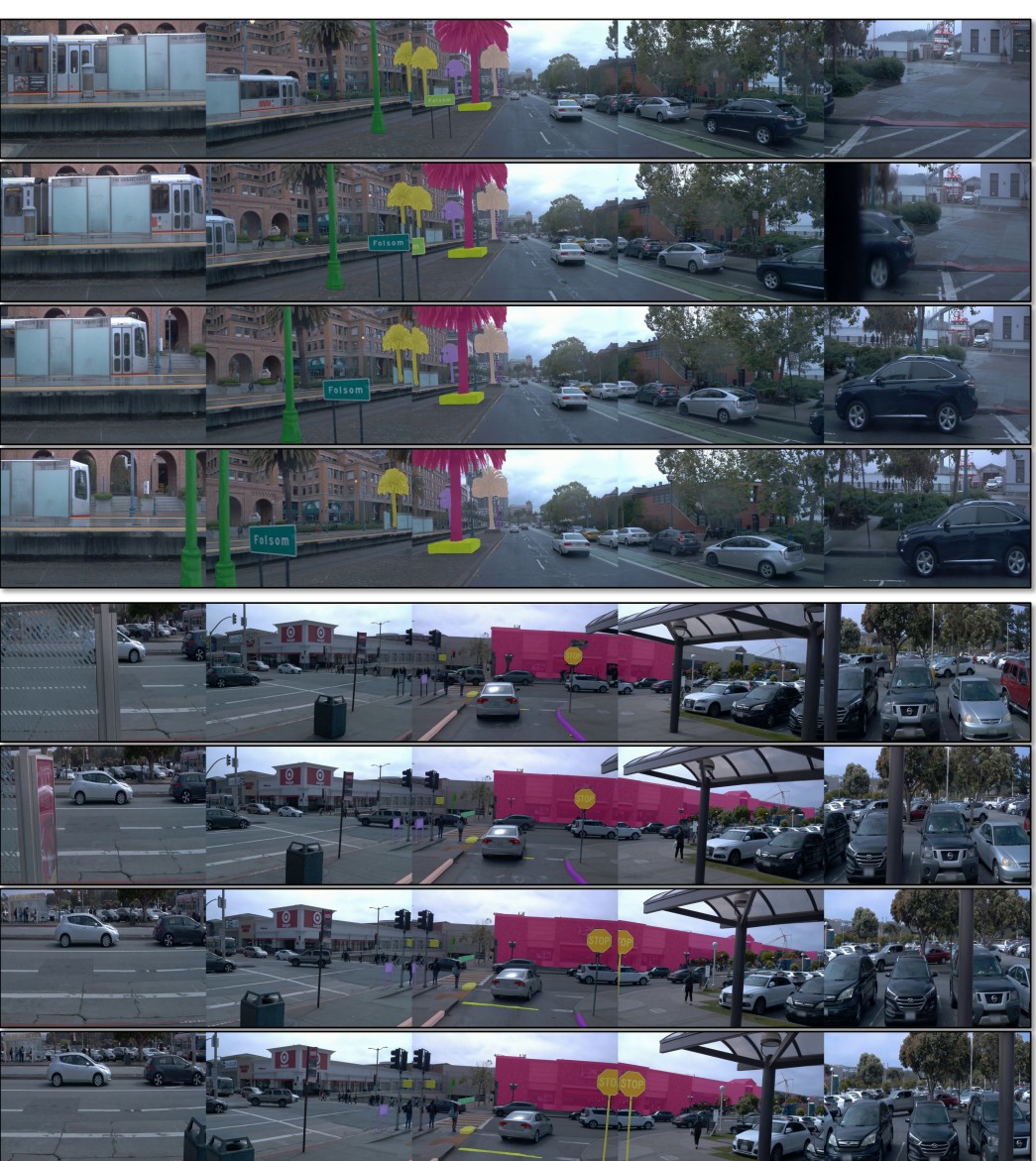

Figure 9: More visualization results of MST-SAM on the Waymo Open dataset. The displayed image has been resized for better visual effects.

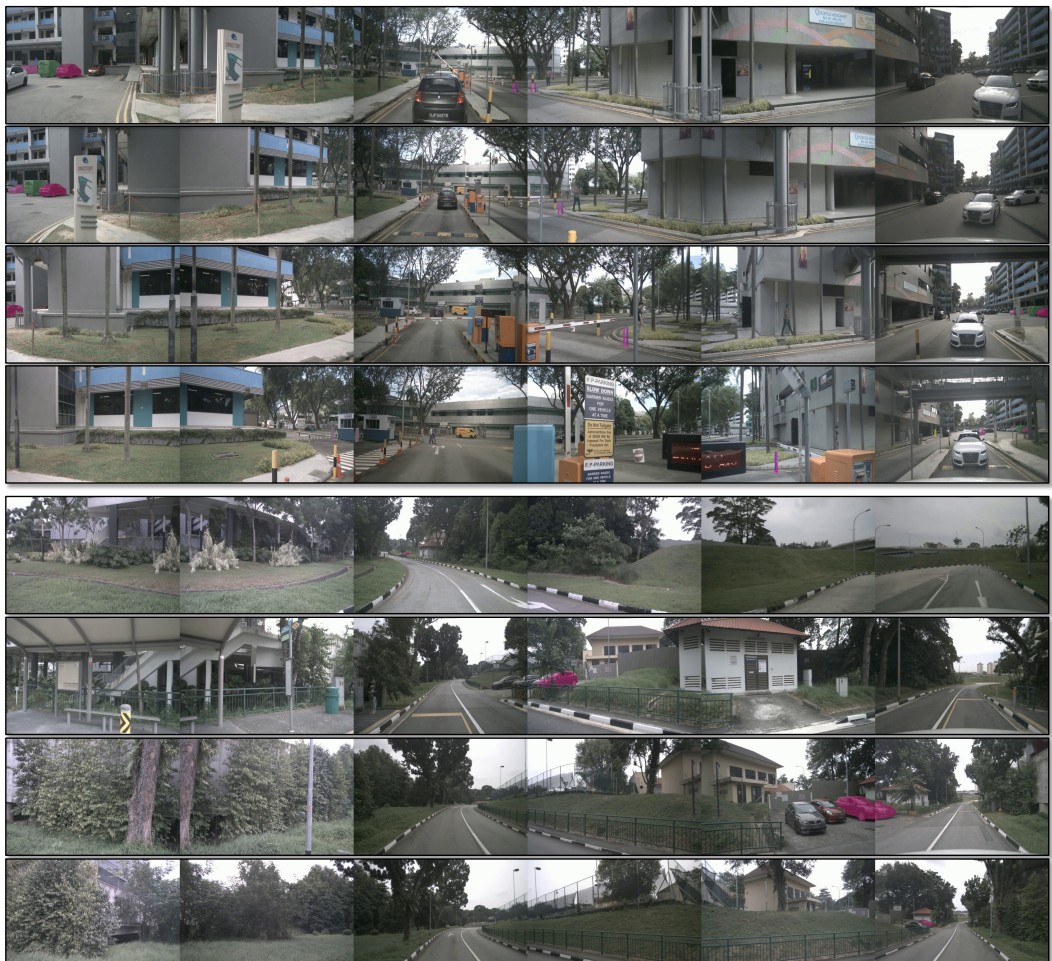

Figure 10: More visualization results of MST-SAM on the nuScenes dataset. The displayed image has been resized for better visual effects.

