# OpenReview forum: "MST-SAM: Bridging Multi-View Gaps in SAM2 with Spatiotemporal Bank"
_ICLR.cc/2026/Conference — ICLR 2026 Conference Withdrawn Submission_

### Official Review · Reviewer_zmUL · 2025-10-27

**Soundness:** 2
**Presentation:** 2
**Contribution:** 2
**Rating:** 4
**Confidence:** 3

**Summary:**

This paper proposes MST-SAM, a method for robust instance segmentation in cross-view videos. Built upon the SAM2 model, which tackles instance segmentation for a monocular video, MST-SAM proposes a Spatio-Positional Augmentation (SPA) module to get a positional encoding that combines both 3D geometric encoding and the standard 2D sinusoidal encoding. Besides, it introduceds the Memory View Selection (MVS) strategy, to selectively prune redundant or useless information in the memory bank to avoid the prohibitive memory and computational costs for processing long, multi-view videos. The experiments and ablation study on nuScenes and Waymo show the effectiveness of the whole method and each individual module.

**Strengths:**

1. The method extends the SAM2 method to tackle instance segmentation for multi-view videos and outperforms the naive baseline of serializing the multi-view video into a single pseudo-sequence on two datasets.
2. The method proposes two modules, one utilizes a 3D projection in positional encoding and the other defines a set of rules to make the memory bank more compact.

**Weaknesses:**

1. The writing quality could be improved. Just to name a few places (not an exhaustive list):

(1) L193 and L196 are repeating.

(2) L050 has an incomplete sentence "the original SAM2 framework".

(3) L182, $Fmem,t$ should be $F_{mem, t}$.

2. The geometric positional encoding requires camera intrinsics and poses, which seems to impose more constraints to the data.

(1) I wonder how robust the method is with respect to inaccurate / noisy camera parameters, or maybe poses gotten from running COLMAP.

3. My biggest concern is that the method only did experiments on two autonomous driving datasets. I am curious if the method also works on other kinds of datasets, e.g. indoor ones.

4. I am also curious to see more analysis on the capacity of the method, e.g. is there any constraints on the video lengtht that the method can handle, is there any constraints on the object size, how well the method handles repeating objects e.g. trees.

**Questions:**

1. For the aggregation in equation (4), why do you use max-pooling? Have you experimented with other aggregation e.g. averaging?

2. How many frames of the videos can the method handle?

---

### Official Review · Reviewer_6brn · 2025-10-31

**Soundness:** 3
**Presentation:** 3
**Contribution:** 2
**Rating:** 4
**Confidence:** 4

**Summary:**

This paper presents MST-SAM, an online framework designed to extend SAM2 for multi-view instance segmentation and tracking in autonomous driving scenarios. The method introduces a Spatio-Positional Augmentation (SPA) module to incorporate geometric priors and a Memory View Selection (MVS) strategy to improve efficiency without sacrificing accuracy. Experiments on nuScenes and Waymo show significant performance gains over SAM2 baselines with minimal latency overhead. Overall, the work provides a strong step toward scalable multi-camera segmentation.

**Strengths:**

1. The authors reasonably design a memory-based framework that effectively integrates the Segment Anything Model (SAM) for multi-view, instance-level segmentation and tracking.
2. The proposed method demonstrates significant performance improvements over the baseline, establishing state-of-the-art results on the introduced benchmarks.
3. The paper also reports a competitive latency, with only marginal computational overhead compared to the SAM2-MV baseline, indicating the method’s efficiency and potential applicability in real-time systems.

**Weaknesses:**

1. The comparison set is limited - the proposed method is mainly compared to SAM-based variants. To strengthen the evaluation, it would be beneficial to include comparisons with state-of-the-art video instance segmentation (VIS) methods, which already provide robust temporal object matching.
2. The multi-view object matching process appears to rely on additional mechanisms, but the robustness of temporal matching from recent VIS methods could serve as a valuable baseline reference.
3. Since the paper targets autonomous driving scenarios, the authors should also report inference time and memory consumption comparison to the VIS methods as well.
4. While MST-SAM effectively extends SAM2 with spatio-temporal and geometric modules, the core contributions (the SPA and MVS modules) are incremental rather than conceptually groundbreaking. The method mainly builds upon existing ideas from 3D-aware positional encodings and memory pruning strategies, which may limit its originality.
5. (Minor) Typographical error: On page 5, line 255, the notation T_ego(t_{src}←cam(t_{src}) should be corrected to T_ego(t_{src})←cam(t_{src}) for clarity and mathematical consistency.

**Questions:**

Could the authors clarify the scalability of MST-SAM to larger multi-camera systems or higher frame rates? For instance, how would performance and latency change if the number of cameras or the sequence length increases significantly?

---

### Official Review · Reviewer_25pp · 2025-10-31

**Soundness:** 3
**Presentation:** 1
**Contribution:** 2
**Rating:** 2
**Confidence:** 4

**Summary:**

This paper proposes to augment SAM2 to segment objects in multi-view video streams. This is made difficult by the fact that SAM2 was only trained on single view videos. The authors propose a spatio-positional encoding module, which lifts patches to 3D, and an attention mechanism that performs token selection across views.

**Strengths:**

The paper focuses on a relatively understudied aspect of 3D video segmentation, which is fusing multiple cameras/views. The proposal seems sensible and draws on the strengths of SAM2, rather than inventing a whole new architecture which needs to be trained from scratch. The experiments, while small, seem fairly well done, and cover generalization across views and objects.

**Weaknesses:**

The main issues of the paper are the presentation, novelty, inadequate related work, and a simple missing baseline that may affect the conclusion.

The novelty is always hard to quantify, but in this case it mostly amounts to the application, which is to focus on adding multi-view fusion capabilities to an existing strong video segmentation model. The technical innovation to achieve this is not very large, as it consists of combining elements that are very common in 3D segmentation. For example, the following papers are representative examples that do similar types of 3D segmentation lifting:

- Bhalgat et al., "3D-Aware Instance Segmentation and Tracking in Egocentric Videos", ACCV 2024
- Gu et al., "EgoLifter: Open-World 3D Segmentation for Egocentric Perception", ECCV 2024
- Siddiqui et al., "Panoptic Lifting for 3D Scene Understanding With Neural Fields", CVPR 2023

These and more would be needed to complement the related work, which is currently sparse. The memory attention is, of course, based on very common attention operators (indeed part of SAM2). The memory view selection strategy is heuristic, although it could be somewhat novel, it consists of simple rules. Overall this is not to say that there is no novelty -- the combined system seems novel. It is mostly that the new additions are heavily inspired by very common operations and may not clear the bar for publication.

Another problem is the presentation. As is, the paper is somewhat hard to follow, due to uninformative and distracting phrasings that have clear LLM-generic-text markings (everything is "critical", "elegant", "intelligent", "inherent", etc), some errors (e.g. repetition of a phrase in lines 192 and 195), wrong use of bibtex's citet instead of citep, etc. I hope that the heavy use of LLMs is disclosed. Overall the manuscript needs more human proofreading to improve the clarity of presentation.

Finally, there is one missing baseline that is very important: fine tuning SAM2 on the same dataset (e.g. Waymo). Since the original SAM2 was not fine-tuned in the same data distribution, we do not know to what degree the proposals improve performance, vs. just fine-tuning on the same data.

**Questions:**

I would like to ask the authors about the baseline - if it is in the paper, I missed it.

---

### Official Review · Reviewer_xfN1 · 2025-11-01

**Soundness:** 2
**Presentation:** 2
**Contribution:** 2
**Rating:** 2
**Confidence:** 4

**Summary:**

This work targets multi-view + temporal instance segmentation for multi-camera systems.

They introduce two ideas to extend SAM2 to this setting:
(1) Spatio-Positional Augmentation (SPA): Uses camera intrinsics and extrinsics to encode a ray, concatenates the ray with depth information, then max pools to get 3D positional embeddings.
(2) Memory View Selection (MVS): a heuristic based strategy for managing temporal context.

For training, they use hybrid sampling (80% cross-view, 20% single-view instances) on nuScenes and Waymo.
They introduce a custom cross-view benchmark on these datasets, measuring J (region IoU) and F (boundary F-measure).
They show significant improvement over FastPoly-SAM.

**Strengths:**

The method section is written clearly.

The mathematical notation is concise and easy to follow.
Figures 3 and 4 are illustrative and explain the ideas quite well.

**Weaknesses:**

The paper doesn't discuss/compare against much of the multi-view perception literature that already have established ways to handle 3D positional embeddings.
The 3D positional embedding approach very closely resembles the ideas proposed from PETR/PETRV2 (Liu et al. ECCV 2022, ICCV 2023).
Even though the tasks differ (BEV detection vs multi-view segmentation), these methods must be in the related work and considered for comparison.

The baselines they compare against are primarily ablations of their own architecture design(MST-SAM-L, MST-SAM-G, MST-SAM-M) rather than published methods.
Only FastPoly-SAM is an external baseline, making it difficult to assess performance against the broader literature.

**Questions:**

The SPA module uses max pooling to aggregate the discretized ray's features.
I don't quite see the intuition behind this - could the authors elaborate on this?

Why is a new benchmark necessary?
To my understanding the datasets already have established instance segmentation / tracking benchmarks.

---

### Note · Authors · 2025-11-14

I have read and agree with the venue's withdrawal policy on behalf of myself and my co-authors.